# Metformin, Empagliflozin, and Their Combination Modulate Ex-Vivo Macrophage Inflammatory Gene Expression

**DOI:** 10.3390/ijms24054785

**Published:** 2023-03-01

**Authors:** Adittya Arefin, Matthew C. Gage

**Affiliations:** 1Wolfson Institute for Biomedical Research, Division of Medicine, University College London, Gower Street, London WC1E 6BT, UK; 2Department of Comparative Biomedical Sciences, Royal Veterinary College, 4 Royal College Street, London NW1 0TU, UK

**Keywords:** macrophage, diabetes, inflammation, metformin, empagliflozin, combinations, anti-diabetes drugs

## Abstract

Type-2 Diabetes Mellitus is a complex, chronic illness characterized by persistent high blood glucose levels. Patients can be prescribed anti-diabetes drugs as single agents or in combination depending on the severity of their condition. Metformin and empagliflozin are two commonly prescribed anti-diabetes drugs which reduce hyperglycemia, however their direct effects on macrophage inflammatory responses alone or in combination are unreported. Here, we show that metformin and empagliflozin elicit proinflammatory responses on mouse bone-marrow-derived macrophages with single agent challenge, which are modulated when added in combination. In silico docking experiments suggested that empagliflozin can interact with both TLR2 and DECTIN1 receptors, and we observed that both empagliflozin and metformin increase expression of *Tlr2* and *Clec7a*. Thus, findings from this study suggest that metformin and empagliflozin as single agents or in combination can directly modulate inflammatory gene expression in macrophages and upregulate the expression of their receptors.

## 1. Introduction

Type-2 Diabetes Mellitus (T2DM) is a complex, chronic illness characterized by persistent high blood glucose levels [1]. In 2017, 425 million people were reported to be suffering from T2DM, with this number projected to rise by 48% by the year 2045 to 629 million [2]. The global yearly expenditure for healthcare costs of diabetes is projected to rise from 727 billion USD (2017) to 778 billion USD (2045) [2].

Acute complications of T2DM include hypoglycemia, diabetic ketoacidosis, and hyperglycemic hyperosmolar nonketotic coma [3,4]. T2DM is strongly correlated with microvascular complications (including diabetic retinopathy, neuropathy, and nephropathy) and macrovascular complications (such as cardiovascular diseases), which are the most common comorbidity associated with T2DM [5]. Intense management of blood glucose levels has been shown to reduce the microvascular complications associated with T2DM [6,7], but its impact on the outcome of cardiovascular diseases such as atherosclerosis is less clear [6,8].

T2DM is a metabolic disease primarily characterized by decreasing sensitivity of cells in the body towards the endogenous insulin (insulin resistance) and decreasing insulin secretion [3], resulting in hyperglycemia. Reduced insulin response may be due to a variety of factors, including lipotoxicity, mitochondrial dysfunction, ER stress, hyperglycemia, and inflammation [9].

### 1.1. Macrophages Play a Significant Role in T2DM Progression

Macrophages are monocyte-derived phagocytic leukocytes of the innate immune system that are commonly associated with response to infection and play important homeostatic roles in angiogenesis and tissue repair. Macrophages also play a central role in the progression of T2DM through their ability to affect insulin response on metabolic tissues, such as liver, muscle, and adipose, through local inflammatory cytokine secretion activating JNK signaling pathways, causing aberrant phosphorylation of insulin receptor substrate proteins [10].

Depending on the tissue microenvironment, monocytes can differentiate into macrophages and have historically been described to polarize into proinflammatory (M1/classical) or anti-inflammatory (M2/alternative) macrophages, though more recent literature demonstrates how macrophage subsets can exist on a spectrum between these two extremes [11,12,13,14,15]. Recent studies have demonstrated that obesity and hyperglycemia promote myelopoiesis in mice and cause an expansion in the pool of circulating classical monocytes [16,17]. Classical short-lived monocytes produce inflammatory cytokines, and these monocytes selectively penetrate the inflamed tissues [11,12,13,14,15]. This metabolic inflammation has become a major focus of research linking obesity, insulin resistance, and T2DM [18], and is characterized by increased immune cell infiltration into tissues, inflammatory pathway activation in tissue parenchyma, and altered circulating cytokine profiles. TNFα, IL1β, IFNγ, and IL6 are major inflammatory cytokines, which are upregulated in diabetes [19] and atherosclerosis [20], and are expressed in macrophages [21].

### 1.2. Treating Patients with T2DM

The management of T2DM is complex due to the chronic nature of the disease, often progressing over decades and integrating the management and treatment of its associated comorbidities [22]. Patients are advised to partake in lifestyle modifications, including maintaining a healthy diet, regular physical activity, and weight-loss [23]. Unfortunately, this is often ineffective [22], and so patients are then prescribed different classes of anti-diabetes agents depending on their blood glucose levels and glycosylated hemoglobin level (% HbA1c) [24].

Common anti-diabetes drugs are aimed at reducing the hyperglycemia [2,25,26], by targeting tissues which directly impact blood glucose levels, for example metformin targets the liver by reducing hepatic glucose output [25] and empagliflozin blocks glucose reabsorption from the kidneys [25]. The availability of different drugs to control hyperglycemia provides ample opportunities for tailoring the treatment regimen according to the individual need of the patient. Typically, patients may be prescribed a single drug or a combination of drugs depending on the severity of their disease [24,25,26], in accordance with health research association guidelines such as the National Institute of Health Care Excellence (NICE) or American Diabetic Association (ADA). This approach imparts an increasing therapeutic burden on the patient, either in the form of dosage upregulation or additional medications [27,28].

The administration of long-term drugs is not without risks [29]. These agents may reduce insulin resistance and increase insulin secretion and glucose absorption from blood [30,31]. However, many of these agents may worsen the co-morbid metabolic disorders in T2DM patients [25,28,30,31]. For example, Thiazolidinediones are potent anti-hyperglycemic agents, yet they have been associated with worsening cardiovascular disease (CVD) and related mortality [32]. Insulin secretagogues, for example sulfonylureas, meglitinides, and DPP-4 inhibitors, have also been associated with higher CVD risk [33,34,35,36].

Recently, the use of anti-inflammatory agents has shown improvement in hyperglycemia control in T2DM patients and disease models [18,37]. Two common features of all of these agents are persistent reduction of inflammation (reduction in CRP levels in blood) and reservation of beta cell function, which collectively resulted in better hyperglycemia management [38,39,40,41,42,43,44,45,46,47,48,49,50,51,52,53,54]. Thus, investigation of how immune cells such as macrophages respond to anti-diabetes agents requires closer attention. Further knowledge of any advantageous or disadvantageous effects of these drugs on the immune system can be utilized to better treat T2DM patients.

### 1.3. Metformin and Empagliflozin Can Affect Macrophages Responses

Several oral anti-diabetic agents have been reported to modulate macrophage polarization towards the M2 anti-inflammatory phenotype, including metformin and empagliflozin [55,56,57]. However, the mechanisms underlying these effects are still poorly understood and may conflict. Metformin has been reported to promote M2 polarization [58] and antitumor or anti-angiogenic M1 polarization [59]. It has previously been shown in murine bone marrow-derived macrophages (BMDM) that lipopolysaccharide (LPS) stimulated phosphorylation of p65 and JNK1 was decreased by metformin, leading to reduced pro-inflammatory cytokine levels [60]. In LPS-stimulated macrophages, the reduction of ApoE expression has been reported to have been reversed by metformin via retarding nuclear translocation of NF-κB [61]. It has also been reported that metformin can inhibit IL1β-stimulated release of IL6 and IL8 from macrophages, human smooth muscle cells, and endothelial cells in a dose-dependent manner [62,63]. 

It has been recently suggested that the cardio-protective activity of empagliflozin [63] may be due to its anti-inflammatory effect [56]. For example, empagliflozin has been reported to reduce the levels of C reactive protein and polarize macrophages towards the M2 phenotype in patients [56,57]. Empagliflozin reduces obesity-induced inflammation via polarizing M2 macrophages in white adipose tissue and liver [64], and empagliflozin has been reported to decrease M1 macrophages and increase M2 in macrophages in the liver and epididymal white adipose tissue of mice [65]. In ex vivo experiments with macrophages stimulated with ATP, it has been observed that empagliflozin can attenuate NLRP3 activation [66]. 

It has been speculated that combining metformin with other drugs with anti-inflammatory effects on the macrophages (e.g., empagliflozin) may help to strengthen the therapeutic potential of metformin [67]. However, while this combination remains to be investigated, it has been previously reported that drug combinations can enhance the anti-inflammatory and anti-oxidant activities in stimulated macrophages [68], and the combination of empagliflozin and gemigliptin has been seen to exert anti-inflammatory activity on LPS-stimulated macrophages [69]. In this investigation, we sought to define the direct immunomodulatory properties of metformin and empagliflozin on macrophages as single agents or in combination, reflecting a clinical approach to patient treatment.

## 2. Results

### 2.1. Metformin Promotes Tnfa and Il1b Inflammatory Gene Expression in Macrophages

To explore the direct effects of metformin on inflammatory gene expression in macrophages, we examined mRNA expression of four well-established inflammatory genes (*Tnfa, Il1b, Il6* and *Ifng*) in mouse BMDM at physiologically relevant concentrations of 1 µM and 10 µM [70,71] at 2 h and 24-h timepoints. We observed that metformin increased mRNA expression of *Tnfa* after 2 h at 1 µM (Figure 1A, 1.41-fold, *p* = 0.002) and 10 µM (Figure 1A, 1.36-fold, *p* = 0.002) and *Il1b* after 24 h (Figure 1F, 6.2-fold, *p* = 0.031).

### 2.2. Empagliflozin Promotes Tnfa, Il1b, Il6, and Ifng Inflammatory Gene Expression in Macrophages 

To explore the direct effects of empagliflozin on inflammatory gene expression in macrophages, we examined mRNA expression of the same four inflammatory genes at identical physiologically relevant concentrations [72] and timepoints. We observed that empagliflozin increased mRNA expression of *Tnfa* after 2 h at 1 µM (Figure 2A, 1.7-fold, *p* = 0.031), *Il1b* at 10 µM after 24 h (Figure 2F, 5.8-fold, *p* = 0.016), *Il6* at 1 µM (Figure 2C, 13.7-fold, *p* = 0.037), and *Ifng* at 10 µM (Figure 2D, 4.5-fold, *p* = 0.011) after 2 h.

### 2.3. Metformin and Empagliflozin in Combination have Contrasting Effects on Macrophage Inflammatory Gene Expression

As metformin and empagliflozin are commonly prescribed in combination, we decided to investigate how the combination of these drugs might compare to the responses observed in the BMDM when they were added as single agents. We observed that in contrast to single drug responses, the combination of metformin and empagliflozin had no effect on mRNA expression of *Tnfa* at 2 h at 10 µM (Figure 3A), however after 24 h incubation, the levels of *Tnfa* mRNA expression were significantly increased (Figure 3E, 1.4-fold, *p* = 0.019). The combination of metformin and empagliflozin reduced mRNA expression of *Il1b* after 24 h (Figure 3F) and *Il6* after 24 h (Figure 3G) when compared to single agent responses (Figure 1 and Figure 2).

### 2.4. In Silico Docking of Empagliflozin with TLR2 and DECTIN1

The direct effects of metformin and empagliflozin on basal macrophage gene expression have not been reported previously. Inflammatory gene expression in macrophages can be induced through the macrophage’s expression of pathogen-associated molecular pattern (PAMP) recognition receptors, which include the toll-like receptors (TLRs) [73] and DECTIN1 [74]. Therefore, we speculated that the proinflammatory signaling we observed may be induced through these receptors. When investigating the structure of empagliflozin (PubChem CID: 11949646), we noticed that empagliflozin has a similar moiety to yeast zymosan (PubChem CID: 64689) (Figure 4B). Zymosan is a well-established activator of inflammatory gene expression in macrophages through TLR2 and DECTIN1 [75,76,77].

In silico protein–ligand docking assessment suggests that both zymosan (Figure 4A) and empagliflozin (Figure 4C) could interact with the TLR2 through hydrogen bond interactions with amino acid residues R423, V425, D444, S445, and S447 (Figure 4). Remarkably, despite having multiple H-bond donor and acceptor groups, the H-bond formation between the residues of TLR2 and empagliflozin seemed to be facilitated only by the moiety identical to zymosan (Figure 4 and Table 1) with better predicted binding energy (−6.0 kcal/mol) than zymosan (−4.2 kcal/mol) (Table 1).

A similar result was observed during docking simulations with DECTIN1-Zymosan and DECTIN1-empagliflozin. Zymosan (Figure 5A) can interact with DECTIN1 receptor through H-bond formation with H126, K128, S129, Y131, N159, and E241 amino acid residues. On the other hand, empagliflozin can form H-bonds with Y131 and N159 amino acid residues of DECTIN1 (Figure 5C). Again, the interaction of empagliflozin with DECTIN1 seems to be facilitated by the moiety identical to zymosan (Figure 5 and Table 2) and yields better binding energy (−6.1 kcal/mol) than zymosan (−5.0 kcal/mol) (Table 2). 

### 2.5. Metformin and Empagliflozin can Interact with Tlr2 and Clec7a and Modulate Their Expression

Follow-up experiments investigating the effects of metformin and empagliflozin either as single agents or in combination with *Tlr2* and *Clec7a* (the gene symbol for DECTIN1) expression revealed that empagliflozin and metformin added as single agents at 10 µM increase *Tlr2* expression (Figure 6A,C) at 2 h (1.53-fold, *p* = 0.0002; 1.38-fold, *p* = 0.003) and 24-h timepoints (1.37-fold, *p* = <0.0001; 1.26-fold, *p* = 0.0005), respectively. However, in combination, Tlr2 expression was less elevated (Figure 6A,1.24-fold, *p* = 0.045) or negated (Figure 6C). Interestingly, this mirrors the expression pattern of *Tnfa* after 2-h exposure (Figure 3A). Regarding *Clec7a* expression, exposures of 10 µM metformin or 10 µM empagliflozin also showed a trend towards increased *Clec7a* expression (Figure 6D) at 24-h (2.33-fold, *p*= 0.06; 2.23-fold, *p*= 0.08), respectively. However, at the 2 h time point tested (Figure 6B), metformin, empagliflozin, and their combination reduced *Clec7a* expression.

## 3. Discussion

Depending on the severity of their disease, patients with type 2 diabetes may be treated with monotherapy (such as metformin) or dual therapy combinations (such as metformin and empagliflozin combination) [25]. Macrophage-driven inflammation plays a significant role in the progression of T2DM [78] and its associated comorbidities, such as atherosclerosis [79]. While reports are emerging of the indirect effect of anti-diabetes drugs on macrophages through polarization [80], the direct responses of anti-diabetes drugs on these cells have remained unstudied. In this investigation, we sought to determine the direct immunomodulatory properties of two of the most commonly prescribed anti-diabetes drugs, metformin and empagliflozin, on macrophages.

Metformin is a biguanide whose mode of action in reducing blood glucose is through reducing hepatic glucose production. Metformin does not require metabolization for its biological activity [70], and physiological plasma levels for biological activity were reported to be between 1 µM to 40 µM with a half-life of 6.5 h [71]. Empagliflozin is an SGLT2 inhibitor whose mode of action is to block glucose reabsorption in the kidney. The physiological plasma levels for biological activity of empagliflozin varies between 1.87 µM to 4.74 µM based on the administered dosing (10 mg and 25 mg, respectively), and it is excreted from the body in an unchanged form after activity. The half-life of empagliflozin is 12.4 h [72]. Therefore, to ensure the clinically relevancy of our experiments, we used metformin and empagliflozin at 1 µM and 10 µM for 2 h and 24 h to determine their direct immunomodulatory effect on murine bone marrow derive macrophages. Murine BMDM from LdlrKO mice are a well-established model for investigating macrophage responses in a cardiometabolic setting [81,82,83,84]. Exposing BMDM to metformin at 1 µM and 10 µM for 2 h increased the mRNA expression of *Tnfa* (Figure 1A) and 24-h exposure at 10 µM significantly increased the mRNA expression of *Il1b* (Figure 1F). Exposing BMDM to empagliflozin also induced *Tnfa* expression at 1 µM within 2 h (Figure 2A), and *Il1b* mRNA expression was significantly increased after 24 h (Figure 2F). Significant increases in mRNA expression were also observed with *Il6* at 1 µM within 2 h (Figure 2C), and *Ifng* within 2 h at 10 µM (Figure 2D). Therefore, within the first 24 h, after physiologically relevant concentrations of metformin or empagliflozin exposure, several major inflammatory genes were observed to be upregulated.

Tnfa, Il1b, and Il6 are activated through TLR signaling [85]. Therefore, we speculated that the proinflammatory signaling we observed may be induced through these receptors. When investigating the structure of empagliflozin (PubChem CID: 11949646), we noticed that empagliflozin has a similar moiety to yeast zymosan (PubChem CID: 64689) (Figure 4B). Zymosan is a well-established activator of inflammatory gene expression, including *Tnfa* and *Il1b* in macrophages [75,76,77] through toll-like receptor 2 (TLR2) and DECTIN1 (mouse gene symbol *Clec7a*) [74,77], and we speculated that the drug–receptor interaction may be TLR2- and DECTIN1-mediated. To test this hypothesis, in silico molecular docking experiments were performed with crystal structures of TLR2 (Figure 4) and DECTIN1 (Figure 5) and the molecules zymosan and empagliflozin. The docking simulations not only suggested that empagliflozin can interact with both TLR2 and DECTIN1 receptors by similar amino acid residue interactions (Table 1 and Table 2) but also yielded better predicted binding energies for both the receptors compared to zymosan (Table 1 and Table 2). These in silico docking experiments also revealed that only the zymosan-moiety in the empagliflozin chemical structure was predicted to be able to interact with TLR2 (Figure 4B,C) and DECTIN1 (Figure 5B,C) receptor amino acid residues. Collectively, these observations indicate a probable recognition of pathogen-associated molecular pattern (PAMP) in the empagliflozin chemical structure by the macrophages. Ligand–receptor binding often modulates mRNA expression of the receptors involved [86]. Further investigation revealed that empagliflozin modulates *Tlr2* and *Clec7a* mRNA expression (Figure 6) in BMDM within the same timeframes observed for inflammatory gene expression, lending support to their possible interaction. 

Regarding the possible mechanism of metformin’s upregulation of the inflammatory genes observed, there is little in the literature regarding metformin’s direct effect on macrophages. Metformin has historically been characterized by its ability to reduce hepatic glucose production through the transient inhibition of the mitochondrial respiratory chain complex I [70,71] and activation of the cellular metabolic sensor AMPK [87]. Under physiological conditions, metformin exists in a positively charged protonated form, which may rely on different isoforms of the organic cation transporters (OCT) to enter the cell [88,89,90]. However, over the last 15 years, a much more complex picture of metformin’s roles is emerging, reflecting multiple modes of action which have AMPK independent mechanisms, with the new findings varying depending on the dose and duration of metformin used [91]. Our experiments revealed that metformin also upregulated *Tlr2* and *Clec7a* mRNA expression (Figure 6), providing an opportunity for the mechanism behind this observation to follow-ups in future investigations.

TNFα is an early response cytokine secreted by macrophages in response to pathogens, which stimulates an acute phase immune response via pathogen-associated molecular pattern (PAMP) receptors such as Toll like receptor 2 (TLR2) by regulating chemokine release and aiding further immune cell recruitment [92]. In macrophages, the half-life of TNFα is approximately 45 min and at least 30 min for mRNA [93] and protein [94], respectively. Our results suggest that macrophages upregulate *Tnfa* expression after being exposed to single antidiabetic agents (Figure 1A and Figure 2A). A similar increase was also observed after 24-h exposure (Figure 1E), however this did not reach statistical significance, possibly reflecting the more immediate nature of the TNFα response. The difference in effects observed at the higher concentration of 10 µM resembles typical responses observed through PAMP receptor stimulation, whereby higher doses of PAMPs lead to a more intense immune response [95,96]. Like TNFα, IL1β is also a pyrogenic cytokine produced by macrophages to initiate an inflammatory response to stimuli in its microenvironment. IL1β also regulates cytokine release, acting as a chemoattractant for recruitment of immune cells to the site of inflammation [92]. One key difference between the two cytokines is that IL1β is synthesized as a leaderless precursor that must be cleaved by inflammasome-activated caspase-1 and then secreted as a mature isoform [97]. Thus, compared to TNFα secretion and action, IL1β secretion and action become evident at a later time point. Our results demonstrate a similar pattern with exposure to single antidiabetic agents as significant increases in *Il1b* expression are observed at the later 24-h timepoint (Figure 1F and Figure 2F). IL6 is a pleotropic cytokine with both inflammatory [96] and anti-inflammatory [98] effects and shared regulation pathways with TNFα and IL-1β production and secretion [92,99]. It has been previously observed in murine macrophages that TLR2 activation results in NF-κB activation, which leads to an up-regulation of Il6 expression [100]. Our results suggest that the increases we observe in *Il6* mRNA expression (Figure 1C and Figure 2C) may also be TLR2-mediated. IFNγ primes macrophages for enhanced microbial killing and inflammatory activation by TLRs [101,102,103]. In response to classic TLR stimulators (e.g., LPS), macrophages produce IFNγ [104,105]. Our results also suggest simultaneous upregulation of *Ifng* and post TLR-activation *Tnfa* expression [92] (Figure 1A,D and Figure 2A,D). In addition, it has been reported that TLR2 stimulation in macrophages can retard the effects observed at 24-h exposure to IFNγ [106,107]. Observations from our study suggest that post-TLR-activation Tnfa levels remained upregulated at 24-h exposure to the drugs or combination (Figure 1E, Figure 2E, and Figure 3E), and *Tlr2* expression also remained significantly upregulated (Figure 6C), although the previously observed upregulation in *Ifng* expression was lost at 24-h exposure (Figure 1H, Figure 2H, and Figure 3H). Thus, it is possible that the drugs metformin and empagliflozin, alone or in combination, have mounted a potent TLR2-mediated initial response, augmented with upregulated *Ifng* expression.

Our results are in contrast to the majority of studies which report anti-inflammatory properties of metformin [56,61,62,63,68] and empagliflozin [57,65,66,67,108]. However, these studies either report (1) indirect systemic anti-inflammatory effects, which may be due to confounding factors such as reductions in hyperglycemia [56,57,61,63,64,68], or (2) polarizing effects [58,60,64,65,66,87]. 

As metformin and empagliflozin are often administrated in combination [26] to patients with type 2 diabetes, we continued our investigation by exploring the effects of these drugs at 10 μM and at 2 h and 24 h time points. We observed that when added in combination, the pro-inflammatory effects observed with single drug exposure at 2-h were negated (Figure 3A,F). A similar pattern of differential modulation was seen with 24-h exposure for *Tlr2* expression (Figure 6A). The mechanism of these reduced responses with metformin and empagliflozin combination may be due to these drugs being recognized by the same set of pattern recognition receptors and leading to competitive inhibition or development of tolerance due to sequential or simultaneous treatment with multiple or higher doses of PAMP [95].

Surprisingly, the exposure to combination of drugs significantly increased *Tnfa* mRNA expression at 24 h (Figure 3E), and the same combination significantly decreased *Il6* mRNA expression at 24 h (Figure 3G). Our data highlight the complexities of individual-gene macrophage inflammatory response regulation; we showed a clearly coordinated proinflammatory response mediated by several genes to a single agent challenge (Figure 1 and Figure 2), which can be negated (Figure 3A,F) or amplified (Figure 3E) when challenged by a combination of those same agents (Figure 7).

Based on our observations, to discover the exact mode of binding of these drugs to macrophages, further techniques for studying drug–receptor interactions (e.g., X-ray crystallography or surface plasmon resonance) would need to be explored. During the EMPA-REG BASALTM trial (a part of the EMPA-REG OUTCOME trial), it was reported that after therapy with empagliflozin, pancreatic beta cell function and sensitivity to glucose were significantly improved, along with a significant reduction in fasting blood glucose and % HbA1c levels [109,110]. However, these studies attributed these remarkable beneficial effects of empagliflozin to its potency in reducing glucotoxicity [109,110,111] via SGLT-2 inhibition. It has recently been reported that the postprandial phase potentiates macrophage-derived IL-1β production that in turn stimulates insulin secretion, synergistically promoting both glucose disposal and inflammation [112]. From our study, it has become evident that *Il1b* expression in macrophages is significantly upregulated at 24 h exposure to empagliflozin. Thus, there is the possibility that in people with diabetes, empagliflozin can potentiate IL-1 β secretion from macrophages, which may explain the improvement in pancreatic beta cell function and sensitivity to glucose observed in the EMPA-REG BASAL^TM^ trial [109,110,111]. Further studies could be conducted to profile blood-derived macrophages and their IL-1β secretion levels in type 2 diabetes patients being treated with empagliflozin to explore a potential correlation.

## 4. Materials and Methods

### 4.1. Animal Work and Cell Culture

All animal procedures and experimentation were approved by the UK’s Home Office under the Animals (Scientific Procedures) Act 1986, PPL 1390 (70/7354). In keeping with previous in vivo cardiometabolic studies [81,82,83,84], BMDM were prepared from low-density lipoprotein receptor knock-out mice (LdlrKO) and cultured as described before [113,114]. In brief, L929 Conditioned Medium (LCM) was used as a source of M-CSF for the differentiation of the macrophages. After 6 days of differentiation, LCM-containing medium was removed, and cells were washed three times in warm PBS and incubated in DMEM containing low endotoxin (≤10 EU/mL) 1% FBS and 20 µg/mL gentamycin without any LCM before being treated with anti-diabetes drugs (metformin; Sigma-Aldrich, Gillingham, UK, empagliflozin; Generon, Slough, UK) for the concentrations and durations indicated.

### 4.2. Gene Expression Analysis

Total RNA from BMDM was extracted with TRIzol Reagent (Invitrogen, Loughborough, UK). The sample concentration and purity was determined using a NanoDrop™ 1000 Spectrophotometer and cDNA was synthesized using the qScript cDNA Synthesis Kit (Quantabio, Leicestershire, UK). Specific genes were amplified and quantified by quantitative Real Time-PCR, using the PerfeCTa SYBR Green FastMix (Quantabio, Leicestershire, UK) on an MX3000p system (Agilent, Stockport, UK). Primer sequences are shown in Appendix A. The relative number of mRNAs was calculated using the comparative Ct method and normalized to the expression of cyclophilin.

### 4.3. In Silico Molecular Docking Simulation

A high resolution (2.4 Å) 3D crystal structure of TLR2 (PDB ID: 3A7C) was selected from the protein data bank [115] and converted to PDB format. This structure was then processed to present the proper size, orientation, and rotations of the protein [116]. The processing was carried out in UCSF Chimera (version 1.14) (https://www.cgl.ucsf.edu/chimera/ (accessed on 16 December 2021) to remove non-standard amino acids, water molecules, ligands and ions, add missing hydrogen atoms, and to perform energy minimization of the protein structure [117]. The 3D structures of Zymosan (PubChem CID: 64689) and Empagliflozin (PubChem CID: 11949646) were obtained in sdf format from PubChem [118]. As total equalization of electronegativity of compounds (or ligands) lead to chemically unacceptable predictions, in order to prepare the ligands for docking simulation, partial charges were assigned to each compound following the Gasteiger method [119], followed by energy minimization in UCSF Chimera (version 1.14). After processing, these molecules were saved as ‘mol2’ files for molecular docking. The docking experiments were conducted with processed protein and ligands using PyRx 0.8 docking software [120]. The same process was repeated with a high resolution (2.8 Å) 3D crystal structure of Dectin-1 (PDB ID: 2CL8) to assess probable interaction with Zymosan (PubChem CID: 64689) and Empagliflozin (PubChem CID: 11949646).

### 4.4. Statistical Analysis

Results are expressed as mean ± SEM. Comparisons within groups were made using one-way ANOVA with Dunnett’s correction applied. *p* ≤ 0.05 was considered statistically significant.

## 5. Conclusions

In this investigation, we sought to determine the direct immunomodulatory properties of the two of the most commonly prescribed anti-diabetes drugs: metformin and empagliflozin on macrophages. Murine bone marrow-derived macrophages were exposed to clinically relevant concentrations and durations of metformin or empagliflozin in single doses and in combination. Our data suggest that both metformin and empagliflozin, as single agents, may elicit inflammatory responses in BMDM through cytokine and receptor expression, and these responses are altered when the drugs are added in combination.

## Figures and Tables

**Figure 1 ijms-24-04785-f001:**
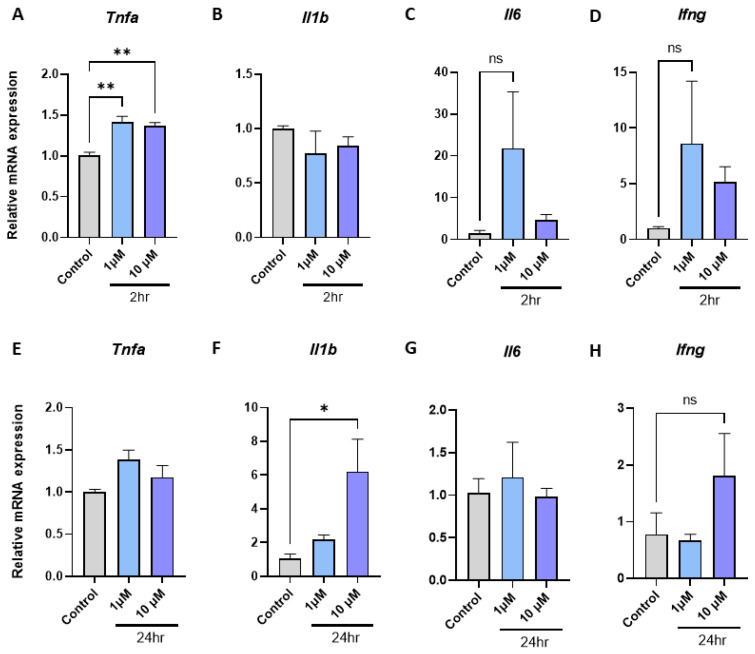
Metformin elicits direct proinflammatory gene expression in BMDM in a time- and dose-dependent manner. (**A**–**D**) Metformin 2 h, (**E**–**H**) metformin 24 h (*n* = 3–4 per group, one-way ANOVA, data are mean ± SEM, * *p* ≤ 0.05, ** *p* ≤ 0.01 relative to control, ns; not significant).

**Figure 2 ijms-24-04785-f002:**
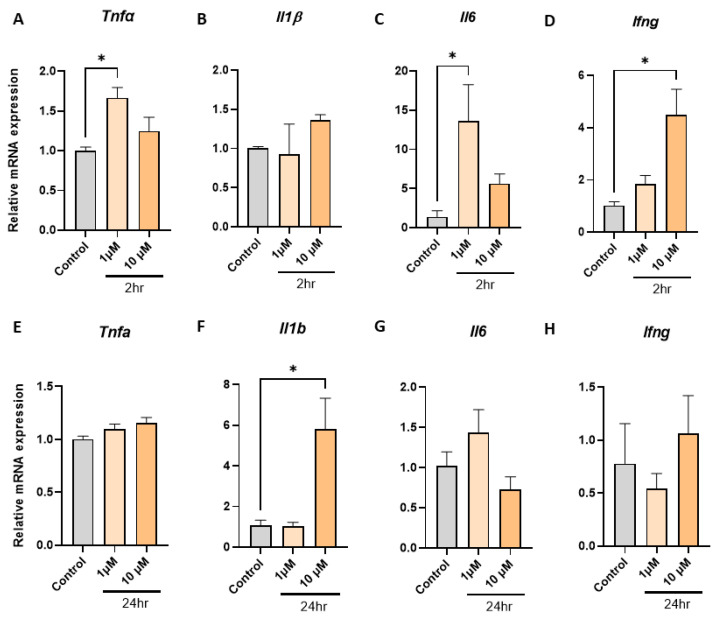
Empagliflozin elicits direct proinflammatory gene expression in BMDM in a time- and dose-dependent manner. (**A**–**D**) Metformin 2 h, (**E**–**H**) metformin 24 h (*n* = 3–4 per group, one-way ANOVA, data are mean ± SEM, * *p* ≤ 0.05 relative to control).

**Figure 3 ijms-24-04785-f003:**
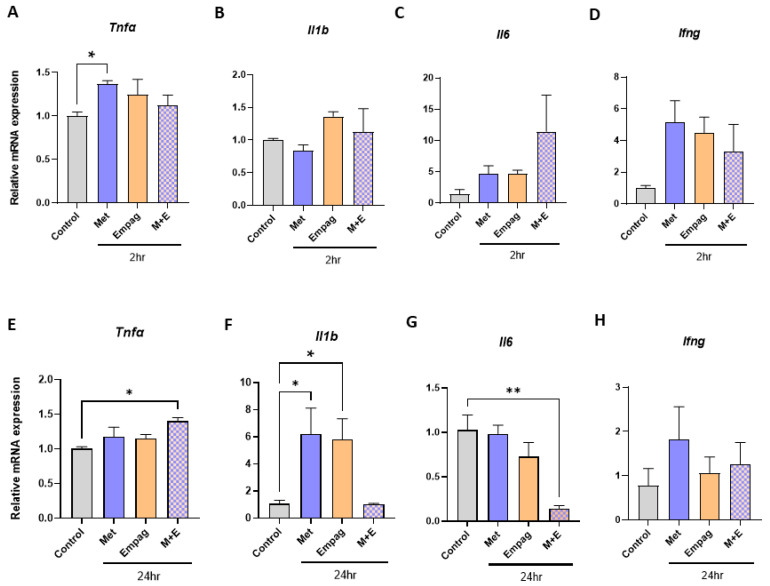
Metformin and empagliflozin in combination have contrasting effects on inflammatory gene expression in BMDM compared to single agents. (**A**–**D**) 2 h, 10 μM, (**E**–**H**) 24 h, 10 μM (Met = Metformin, Empag = Empagliflozin, M+E = combination, *n* = 3–4 per group, one-way ANOVA, data are mean ± SEM, * *p* ≤ 0.05, ** *p* ≤ 0.01 relative to control).

**Figure 4 ijms-24-04785-f004:**
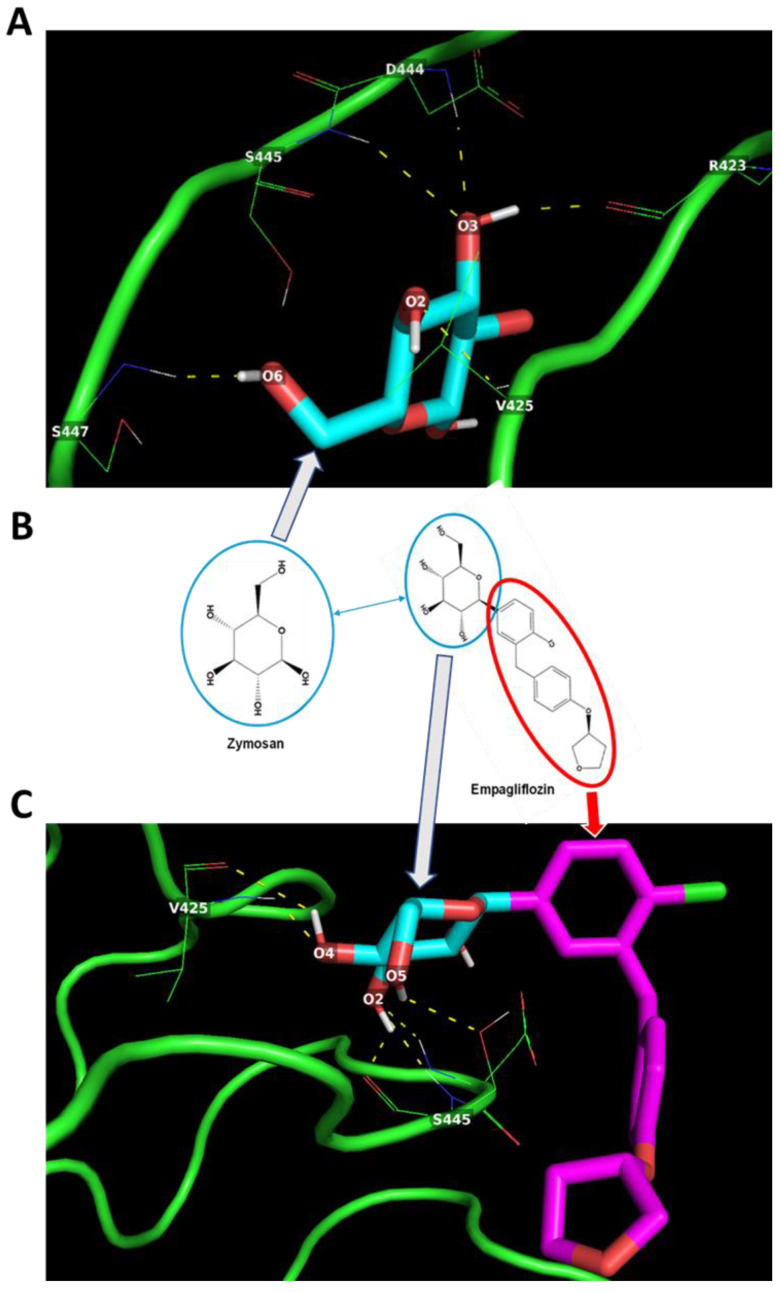
Potential zymosan and empagliflozin interactions with TLR2. (**A**) Zymosan may interact via multiple hydrogen bonds (dotted yellow lines) with R423, V425, D444, S445, and S447 amino acid residues of TLR-2. (**B**) Empagliflozin has a moiety identical to Zymosan. (**C**) The zymosan-like moiety may enable Empagliflozin to interact withV425 and S445 amino acid residues of TLR-2 via H-bond formation.

**Figure 5 ijms-24-04785-f005:**
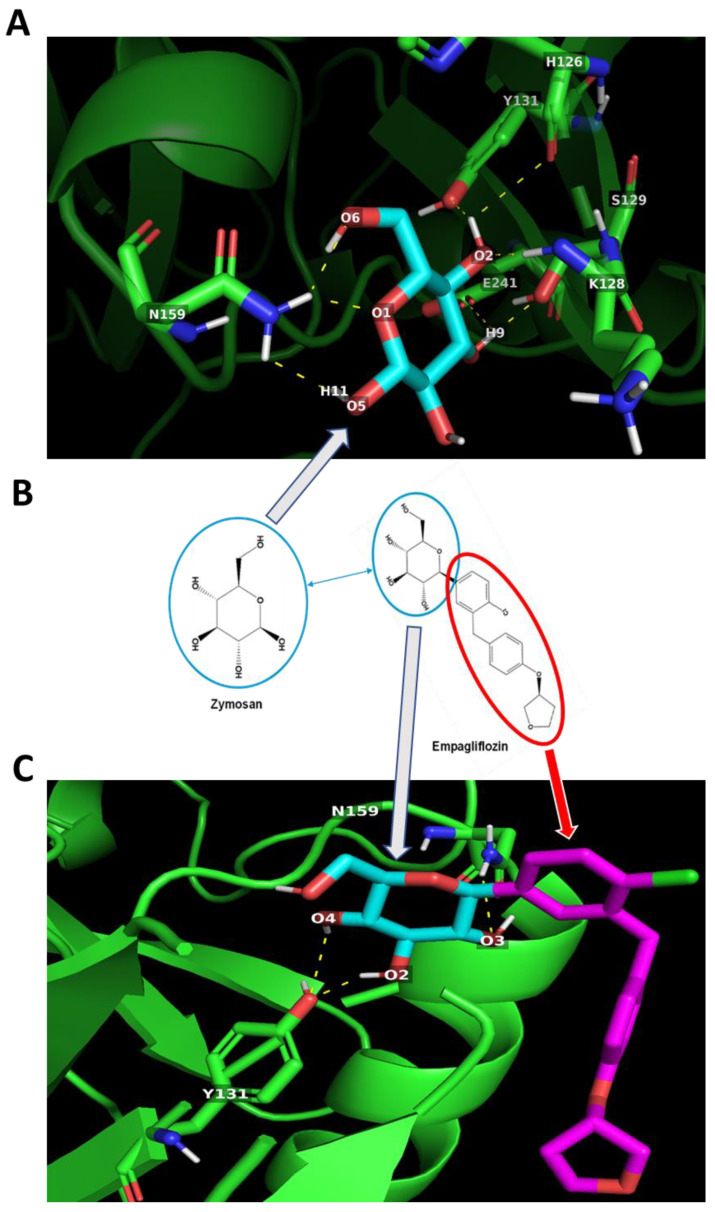
Potential Zymosan and empagliflozin interactions with Dectin-1. (**A**) Zymosan may interact via multiple hydrogen bonds (dotted yellow lines) with H126, K128, S129, Y131, N159, and E241 amino acid residues of Dectin-1. (**B**) Empagliflozin has a moiety identical to Zymosan. (**C**) The Zymosan-like moiety may enable Empagliflozin to interact with Y131 and N159 amino acid residues of Dectin-1 via H-bond formation.

**Figure 6 ijms-24-04785-f006:**
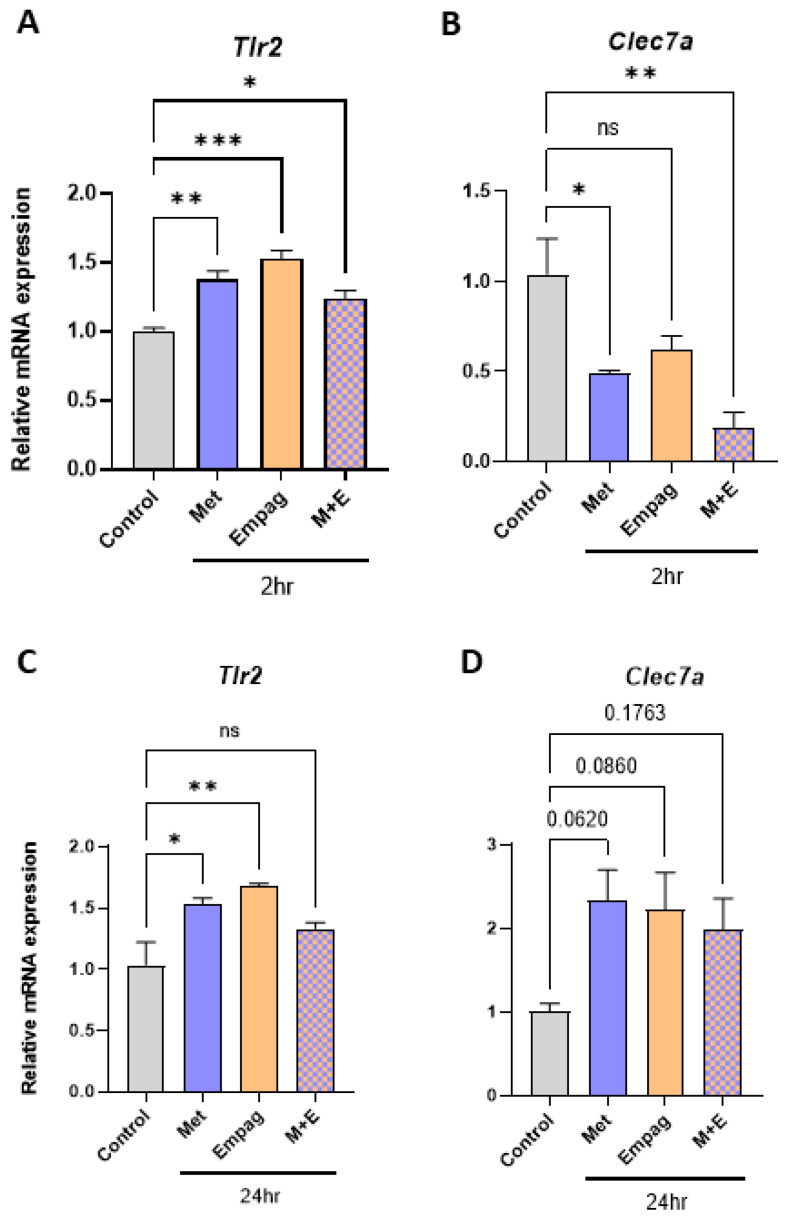
Metformin and empagliflozin as single agents or in combination have contrasting effects on inflammatory gene expression in BMDM compared to single agents. (**A**,**B**) 2 h, 10 μM, (**C,D**) 24 h, 10 μM (Met = Metformin, Empag = Empagliflozin, M+E = combination, *n* = 3–4 per group, one-way ANOVA, data are mean ± SEM, * *p* ≤ 0.05, ** *p* ≤ 0.01, *** *p* ≤ 0.001 relative to control, ns; not significant).

**Figure 7 ijms-24-04785-f007:**
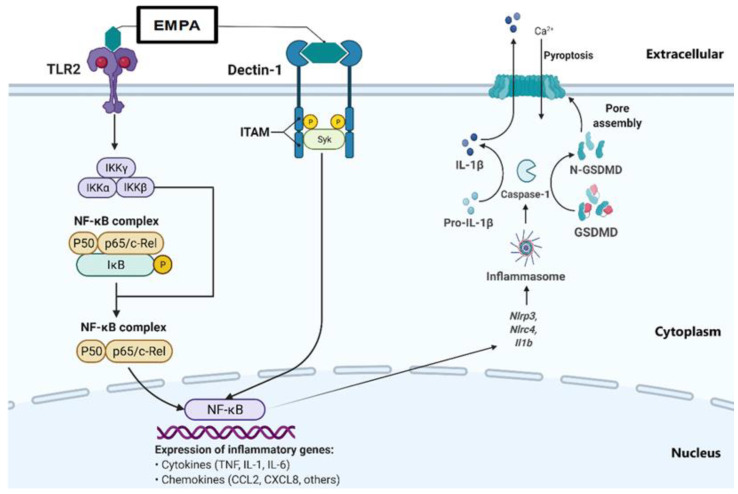
Schematic summarising of the potential interaction of metformin and empagliflozin with TLR2 and Dectin-1 and how they may modulate macrophage inflammatory responses (Empa = empagliflozin, TLR2 = Toll-like receptor-2, Dectin-1 = C-type lectin domain family 7 member A, IKKγ = Inhibitor of nuclear factor kappa-B kinase subunit gamma, IKKα = Inhibitor of nuclear factor kappa-B kinase subunit alpha, IKKβ = Inhibitor of nuclear factor kappa-B kinase subunit beta, NF-κB= Nuclear factor kappa-light-chain-enhancer of activated B cells, ITAM = Immunoreceptor tyrosine-based activation motif, SYK = Spleen tyrosine kinase, TNF= Tumor necrosis factor, IL-1 = Interleukin-1, IL-6 = Interleukin-6, CCL2 = CC chemokine receptor 2, CXCL8 = Chemokine (C-X-C motif) ligand 8, IL-1β = Interleukin-1 beta, Nlrp3 = NOD-, LRR-, and pyrin domain-containing protein 3, Nlrc4 = NLR Family CARD Domain Containing 4, GSDMD = Gasdermin D, N-GSDMD = N-terminal fragment of GSDMD. Created in BioRender.Com.

**Table 1 ijms-24-04785-t001:** Predicted protein–ligand interactions for TLR2-Zymosan and TLR2-Empagliflozin with binding energies from docking simulations.

Target Protein	Ligand	Potential H-Bond Formation	Predicted Amino Acid Residue Interaction (Number)	Predicted Binding Energy (kcal/mol)
TLR2	Zymosan	5	R423 (1) V425 (1) D444 (1) S445 (1) S447 (1)	−4.2
TLR2	Empagliflozin	6	V425 (2) S445 (4)	−6.0

**Table 2 ijms-24-04785-t002:** Predicted protein–ligand interactions for DECTIN1-Zymosan and DECTIN1-Empagliflozin with binding energies from docking simulations.

Target Protein	Ligand	Potential H-Bond Formation	Predicted Amino Acid Residue Interaction (Number)	Predicted Binding Energy (kcal/mol)
DECTIN1	Zymosan	8	H126 (1) K128 (1) S129 (1) Y131 (1) N159 (3) E241 (1)	−5.0
DECTIN1	Empagliflozin	3	Y131 (2) N159 (1)	−6.1

## Data Availability

Data is contained within the article.

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
