# Peer review of "Metformin, Empagliflozin, and Their Combination Modulate Ex-Vivo Macrophage Inflammatory Gene Expression"

_ijms, 2023, doi:10.3390/ijms24054785_

Round 1
Reviewer 1 Report
In the manuscript - Metformin, empagliflozin and their combination modulate ex-vivo macrophage inflammatory gene expression – authors present data from direct interaction of the two anti-diabetic drugs on mouse bone marrow derived macrophages from a mouse line with LdlrKO. Overall, this is a simple report directly describing the effects of the 2 drugs on exvivo mRNA upregulation of 4 proinflammatory cytokines TNFa, IL1B, IL6 and IFNy.
The manuscript is generally well written although with few typos, grammatical errors, and interchangeable ways of spelling “hyperglycaemia”. Please carefully check all the text and be consistent.
The introduction is succinct and with sufficient information, the results are presented briefly and in discussion authors do try to justify the sometimes-contradictory outputs. It is plausible that each of the 4 cytokines is produced at different time within a macrophage and with different half-life, with IL1b inflammasome pathway taking longer. But this could be substantiated with more evidence.
The only mechanism of action is offered by in silico binding simulation but only for empagliflozin, why not also for metformin? How these drugs compare on a structural level? Could this explain some antagonistic effects in the combination treatment?
What is also missing from the current version of the manuscript is how these findings are relevant to the current clinical treatment and what would be further steps.
Major comments:
L49 – one sentence explaining on how macrophages affect insulin responses on metabolic tissues – through which cellular mechanism/pathway. (Reader can follow reference for details but the manuscript should provide sufficient relevant information).
L131 – Ldlr-KO mice – briefly state what is the KO mouse lacking and why it was a good model to be used in this study
L156 – should be explained why all the accumulative charges need to be reduced to zero before the docking simulations.
Separate section 3.4. for in silico docking of empagliflozin with TLR2 and Dectin1, make a new section 3.5 Metformin and empagliflozin can interact with Tlr2 and Clec7a and modulate their expression - otherwise it is misleading, and a reader expects the molecular docking of metformin too, while the second part is lost in this section.
It would be important to see the molecular docking with metformin too, and how the 2 drugs molecular structure is different/similar in a way of binding to TLR2 or DECTIN1, is there any interaction between them?
L347 – is there any information on half-life of TNFa within macrophages?
Future direction section is missing - I would recommend adding a section of how these findings could be taken further. How perhaps these macrophage profiles could be verified in blood-derived macrophages from patients on metformin, empagliflozin or both. What are the molecular differences between these compounds and how this can be further investigated for patients’ benefit? This would be important considering that the direct effects of these drugs on mouse exvivo cells are contrasting from clinical trials. How authors propose the clinical relevancy of these findings for current diabetic therapies.
Minor comments:
Section 1.1 – This paragraph could be expanded (1-2 sentences) to include homeostatic roles of macrophages alongside response to inflammation, and the more recently identified heterogeneity of macrophage subsets beyond the simple M1/M2 polarisation – to make sure that the paper is relevant within the recent literature context.
L67 – then instead of the in “the patients are the prescribed”
Throughout – some acronyms are not explained at first use but later in the text
L139 – “total RNA from” is left mid-sentence
L286-L305, L340 – there are several grammar errors in this section – please check carefully.
Reviewer 2 Report
Remarks to the Author:
Adittya Arefin and Matthew C. Gage showed that Metformin and empagliflozin can upregulate inflammatory responses in macrophages. Only empagliflozin can interact with TLR2 and DECTIN1, which is interesting to know. But through the whole manuscript, they only detected the mRNA level of indicated genes, which can not reflect protein level. Therefore, it is not enough to make a conclusion.
Major points:
1. The mRNA changes of TNFα and IL1β induced by Metformin treatment 2h or 24h are not strong as Fig1A shows. But other proinflammatory genes are similar. Why?
2. Empagliflozin can increase the expression of TNFα and IL6 at 1uM but not 10 um, which is hard to understand.
3. It seems that combination did not enhance the efficacy than single in Fig3?
4. I only can believe you if you show protein level by ELISA after these two drugs treatment.
Round 2
Reviewer 2 Report
My opinion is accept. The author answered my questions.